# Isolation and Identification of *Streptomyces* spp. from Desert and Savanna Soils in Sudan

**DOI:** 10.3390/ijerph17238749

**Published:** 2020-11-25

**Authors:** Mohamed E. Hamid, Adil Mahgoub, Abdulrhman J. O. Babiker, Hussein A. E. Babiker, Mohammed A. I. Holie, Mogahid M. Elhassan, Martin R. P. Joseph

**Affiliations:** 1Department of Clinical Microbiology and Parasitology, College of Medicine, King Khalid University, P.O. Box 641, Abha 61314, Saudi Arabia; amarmartin4u@gmail.com; 2Department of Preventive Medicine, Faculty of Veterinary Medicine, University of Khartoum, Khartoum North 13314, Sudan; adilmahgo333@gmail.com; 3Department of Microbiology, College of Medical Laboratory Science, Alzeim Alazhari University, Khartoum North 12217, Sudan; microedu713@gmail.com; 4Al-Amal National Hospital, Khartoum North 23622, Sudan; abdohu1995@gmail.com; 5Department of Clinical Science, College of Veterinary Medicine, King Faisal University, P.O. Box 400, Al-Ahsa 31982, Saudi Arabia; hbabiker@kfu.edu.sa; 6Department of Clinical Laboratory Science, College of Applied Medical Science, Taibah University, Al-Madinah 13215, Saudi Arabia; mmemam@taibahu.edu.sa

**Keywords:** actinomycetes, ecosystem, biodiversity, phenotypic identification, 16S rRNA gene

## Abstract

The purpose of this study was to investigate streptomycete populations in desert and savanna ecozones in Sudan and to identify species based on 16S rRNA gene sequences. A total of 49 different *Streptomyces* phenotypes (22 from sites representing the desert and semi-desert ecozone; 27 representing the savanna ecozone) have been included in the study. The isolates were characterized phenotypically and confirmed using 16S rRNA gene sequence analysis. The two ecozones showed both similarities and uniqueness in the types of isolates. The shared species were in cluster 1 (*Streptomyces (S.) werraensis)*, cluster 2 (*Streptomyces* sp.), cluster 3 (*S. griseomycini*-like), and cluster 7 (*S. rochei)*. The desert ecozone revealed unique species in cluster 9 (*Streptomyces* sp.) and cluster 10 (*S. griseomycini*). Whereas, the savanna ecozone revealed unique species in cluster 4 (*Streptomyces* sp.), cluster 5 (*S. albogriseolus/ S. griseoincarnatus*), cluster 6 (*S. djakartensis*), and cluster 8 (*Streptomyces* sp.). Streptomycetes are widely distributed in both desert and the savanna ecozones and many of these require full descriptions. Extending knowledge on *Streptomyces* communities and their dynamics in different ecological zones and their potential antibiotic production is needed.

## 1. Introduction

The phylum Actinobacteria (order Actinomycetales) hosts diverse high G+C, Gram-positive bacteria including members of the genus *Streptomyces (S.)* Streptomycetes are Gram-positive spore-forming bacteria which include more than 900 species (https://www.ncbi.nlm.nih.gov/Taxonomy/Browser/wwwtax.cgi?id=1883). *Streptomyces* represents the largest taxonomic group within the Bacteria Domain which is ubiquitously distributed in both aquatic and terrestrial ecosystems [1,2]. They are found in a wide range of habitats but are particularly abundant in soil, representing around 1 to 20% of the total viable count [2,3] and are known to prefer the characteristic earthy smell (geosmin) [4]. Streptomycetes are chemoheteroorganotrophs and are distinguished by their tough, leathery colonies and filamentous growth. They have an important ecological role in the turnover of organic material and are capable of using complex organic materials such as carbon and energy sources in the breakdown of these products in the soil [5].

The microbial communities including streptomycetes are affected by biotic and abiotic factors, notably vegetation, type of soil, and climate [6] Actinomycetes’ diversity, and particularly that of streptomycetes, is widely documented for various reasons including the continued generation of environmental isolates for pharmaceutical, biodegradative, and biotechnological screening. Actinomycetes isolated from soil and related substrates show primary biodegradative activity [5].

Exploring new habitats around the globe is increasingly becoming the focus for discovering new taxa of Actinobacteria, namely streptomycetes, for potential antibacterial, antitumoral, and antifungal activities [7,8]. The habitat of Sudan has remained virtually untouched except for very few studies. Some areas in Sudan were found to be rich sources of microbial biodiversity, holding within them immense novelty and potentiality of identifying new isolates for the production of life-saving drugs such as amphotericin A. Marine habitats of Sudan were found to be a promising source for *Streptomyces* spp. diversity and a potential source of production of secondary metabolites [9]. Two actinomycin D-producing streptomycetes from Sudanese soil have been characterized and were found to be different from previous actinomycin-producing species [10]. The finding of distinct phylogenetic lineages and the variation in the spatial distribution of clones suggests that selection pressures may vary over the soil landscape [11].

This study aimed to isolate and identify streptomycetes from desert and savanna ecozones in Sudan and to find out the species diversity in the two ecozones using 16S rRNA gene analysis.

## 2. Materials and Methods

### 2.1. Study Samples and Geographical Locations

Soil samples were collected from different sites in two ecozones: (1) a desert and semi-desert ecozone, and (2) a low rainfall woodland savanna ecozone (Figure 1), according to the ecological classification by Harrison and Jackson [12]. The soil samples were collected during the dry season from the surface as well as from 5–10 cm deep. From five sections, 10 g within each site were collected with a sterilized spatula. Soils were then merged, sieved, and thoroughly mixed. The composite sample was transferred into sterile plastic bags, labeled, transported to the laboratory, and stored at 4 °C awaiting analysis.

### 2.2. Pretreatment of Soil Samples

The Higher Nitrogen Content (HNC) medium [13] was used as a pretreatment method to assist the extraction and isolation of streptomycetes from soil. HNC medium was prepared and stored at 4 °C. After collection, 0.5 g of each soil sample was added to 50 mL HNC medium in a sterile Erlenmeyer flask. Flasks were placed on a shaker at 42 °C for 1 h. The suspension was left for 5 min and transferred to a clean Falcon tube.

### 2.3. Isolation of Streptomyces spp.

The International Streptomyces Project agar number 2 (ISP2) [14] supplemented with cycloheximide (50 µg/mL), nystatin (40 µg/mL) and nalidixic acid (54.9 µg/mL) and 1 mL vitamin solution (filtered stock solution in 80 mL DW. pH 7: folate, 1 mg [(0.0125 µg/mL); biotin, 1 mg(0.0125 µg/mL); p-aminobenzoic acid, 20 mg(0.25 µg/mL); thiamine, HCl, 100 mg(1.25 µg/mL); pantothenic acid, 120 mg(1.5 µg/mL); riboflavin, 100 mg(1.25 µg/mL); nicotinic acid, 230 mg(2.875 µg/mL); vitamin B12, 10 mg ((0.125 µg/mL)] was used for the isolation and cultivation of *Streptomyces* spp. Subsequent to autoclaving and cooling to 40 °C, the medium was supplemented with cycloheximide (50 µg/mL), nystatin (40 µg/mL) and nalidixic acid (54.9 µg/mL) to inhibit bacterial and fungal contamination.

Samples were diluted as 1:0 (no dilution); 1:5; 1:10; and 1:30. From each dilution, 0.1 mL was evenly spread onto the ISP2 medium using a sterile Drigalski spatula. Inoculated plates were incubated at 28 °C for up to three weeks. Suspected colonies were picked out and streaked onto fresh ISP2 medium to purify streptomycetes colonies [15]. The pure cultures were stored on sterile vials containing 20% glycerol solution at −20 °C for further analysis.

### 2.4. Characterization of Soil Isolates

The predictable *Streptomyces* isolates were studied for morphological and microscopic properties using light microscopy [2]. Strains that showed mutual characteristics were clustered to form preliminary *Streptomyces* color groups for further analysis.

### 2.5. Molecular Identification of Soil Isolates

The 16S rRNA gene amplification and determination of a reasonable size sequence were performed as described by Rainey et al. [16]. Amplification from the genomic DNA samples was done using the eubacterial universal primers 27 F 5′-AGAGTT TGA TCC TGG CTC AG-3′ and 1492R 5^′^-GGT TAC CTT GTT ACG ACT T-3′ [17]. Amplification reactions were performed in a final volume of 20 μL, containing Promega Green Mix. The thermal cycling conditions were as follows: initial denaturation at 94 °C for 5 min; 31 cycles at 95 °C for 30 s, 54 °C for 90 s and 72 °C for 120 s; and a final extension at 72 °C for 5 min. The amplification reaction was performed by Bio-Rad thermal cycler (MyCycler, Bio-Rad, California, CA 94404, USA) and the amplified products were examined by 1% agarose gel electrophoresis. 

For 16S rRNA gene sequencing, the chromosomal DNA was isolated from a 7-day culture using the PEG 200 (polyethylene glycol, Sigma-Aldrich, Munich, Germany). PCR-mediated amplification of the 16S rRNA gene and DNA sequencing were carried out with two universal primers, 27F and 1492R [18]. The ABI 370XL 96-capillary DNA analyzer (Applied Biosystems) and SeqMan software (DNASTAR) were used to analyze and assemble the gene sequences. 

Obtained DNA sequences were first examined and corrected using Chromas (version 2.6.6 (2018), Technelysium Pty Ltd., South Brisbane, Queensland, Australia). These were then aligned with all available 16S rRNA gene sequences of validly described species using the EZbioCloud 16S database (www.ezbiocloud.net) to reveal an approximate phylogenetic relationship. The phylogenetic tree was constructed by the MEGA X program [19], using the Neighbor-joining method with bootstrap values based on 100 replications. 

## 3. Results

### 3.1. Characteristics of the Study Samples

The soil type in the desert (sites 7 and 19) was a Yermosol type [20] with an average annual temperature of 29.7 °C and an average annual rain of 70 mm. Savanna (sites 8, 10, 14, 23, and 29) had the following soil types: Arenosols, which are stabilized sand dunes with silt or clay mainly found in in sites 8, 10, 14, and 23; whereas site 29 (Ad Damazin; 11.7855 °N; 34.3421°E) had Vertisols soil type. Savanna sites had annual temperatures range from 26.0 to 28.5 °C and annual rainfall ranged from 213 mm in Al Fashir (13.6198° N; 25.3549° E) to 713 mm in Ad Damazin (site 29) (Figure 1).

### 3.2. Isolation and Identification of Streptomyces spp.

Different putative *Streptomyces* phenotypes (22 from sites representing the desert and semi-desert ecozone; 27 representing the savanna ecozone) were recovered and characterized. Various bacterial colonies appeared on ISP2 agar after 3 days of aerobic incubation at 28 °C (Figure 2). Separation and purification of colonies by using the pick spot technique on ISP2 agar master plates were used to purify and store colony mass for further analysis.

### 3.3. Molecular Identification of Isolated Streptomycetes

As a result of 16S rRNA gene sequencing, all 49 strains were confirmed as *Streptomyces* spp. based on partial 16S rRNA gene sequencing. Nucleotide sequence data have been deposited in GenBank and corresponding accession numbers are listed in Table 1. Isolate sequences were compared with sequences of *Streptomyces* type strains, and the relationships between the isolates and their closest phylogenetic neighbors are shown in Figure 3. Some sequences formed distinct phylogenetic lines, while others were grouped in clusters in the *Streptomyces* 16S rRNA gene tree.

The investigated ecozones displayed both resemblances and uniqueness in the types of isolates according to clusters and single-membered strains shown in Table 1 and Figure 3. The shared species were in cluster 1 (*S. werraensis*), cluster 2 (*Streptomyces* sp.), cluster 3 (*S. griseomycini*-like), and cluster 7 (*S. rochei*). The desert ecozone revealed unique species in cluster 9 (Streptomyces sp.) and cluster 10 (S. *griseomycini*). Whereas the savanna ecozone revealed unique species in cluster 4 (*Streptomyces* sp.), cluster 5 (*S. albogriseolus*/ *S. griseoincarnatus*), cluster 6 (*S. djakartensis*), and cluster 8 (*Streptomyces* sp.)

## 4. Discussion

Streptomycetes are widely distributed in nature, especially in soils with different structures and chemistry. This study was undertaken to highlight the presence of streptomycetes in two ecozones (ecosystems) and identify isolates to species levels. The results indicated the similarity in some types of the isolates as well as the uniqueness of each ecosystem. These findings emphasize the value for more analysis towards obtaining novel antimicrobial agents from these streptomycetes from either desert or savanna ecozones. Little scientific work has been carried out in this field, notably in the distinctive ecological ecozone of Sudan.

Many sub-Saharan African countries, including Sudan, have considerably diverse soils, such as clay in the east-central area and dunes in the west and north, all with variable climatic conditions [21]. A range of isolates from these soil types with different annual rainfall were noticed in this study (Figure 3, Table 1). *Streptomyces* species in cluster 9 (Streptomyces sp.) and cluster 10 (S. *griseomycini*). were isolated from the desert but not from the savanna. Likewise, the savanna ecozone, supposedly richer soil for agriculture, disclosed distinctive species namely in cluster 4 (*Streptomyces* sp.), cluster 5 (*S. albogriseolus*/ *S. griseoincarnatus*), cluster 6 (*S. djakartensis*), and cluster 8 (*Streptomyces* sp.). Many other single-member clusters have been found in both ecozones, many of them not fully identified as potential novel species giving their significant nucleotide dissimilarity with all known *Streptomyces* spp. Similar results were obtained from desert and semi-desert soils in Israel, where the differences observed were related to specific environmental factors rather than geographic distances and spatial distribution patterns [22].

It has been shown that numerical taxonomy is of established value both for the circumscription and identification of *Streptomyces* species [23,24]. This method was based on identifying various phenotypic characteristics of organism and the resultant data was analyzed by conventional statistics. It is true, however, that analyzing many phenotypic characteristics is neither easy to perform nor accurate in establishing identification to species level because biological similarities between species in a genus of over 700 members are high [2]. The application of the polyphasic taxonomical approach has been applied and successful, especially with actinomycetes. Even so, it is a labor- and time-intensive approach applying a plethora of techniques including traditional morphological and biochemical tests, DNA–DNA hybridization, and variation in 16S rRNA gene sequence [25]. Our approach in the present study was to study selected cultural and morphological characteristics and to confirm this initial identification with the sequencing analysis of the 16S rRNA gene. This attempt has been successful and saved a lot of time and money. However, not all isolates yielded the complete sequences or had poor endings and beginnings. Therefore, it remains one of the deficiencies of this study that the clustering and the phylogenetic tree was established based on partial gene sequencing for most of the isolates. Although dendrograms generated using both partial (ca. 500 bp) and complete (1500 bp) lengths, results were found to be basically the same with either length [26].

Different clades of *Streptomyces* spp. were found and different strains were isolated, showing a widespread range of antibacterials and differing modes of action [27]. The decrease in the average soil relative humidity and the increase in temperature explain significant reductions in the diversity and connectivity of these desert soil microbial communities and lead to significant reductions in the abundance of key taxa typically associated with fertile soils [28]. Such associations between the abundance of species and soil relative humidity of temperature has not been investigated in the present study and should be taken into consideration in future research. *Streptomyces* species are not just free-living soil organisms; they have a valuable ecological function in the soil as they have evolved to exist in symbiosis with plants, fungi, and animals [29]. Streptomycetes are recognized to yield many secondary metabolites that can reduce the growth of pathogens, including plant pathogens [30,31] and human pathogens [32,33,34]. It is probable that these antibiotics evolved as a direct result of their interactions with other organisms namely in symbiosis with fungi, plants, and animals [29,35]. Such interactions can be parasitic in the same way as the scab-causing streptomycetes [36] and those which infect humans [37,38]. The results of our present study can be a platform for exploiting this newly emerging field of research in which streptomycetes play influential roles in bio-control measures from these unusual and undiscovered fields and geographical locations.

The limitations of the present study include the fact that the dendrogram and the similarity matrices were established using partial lengths of the 16S rRNA gene in most of the strains. The use of almost complete sequences or the application of Multi-Locus Sequence Typing (MLST) will be needed for better descriptions of the unknown species. Microbial ecology and ecological characterization or metabolic characterization of isolated *Streptomyces* has not been conducted. Besides, screening for antimicrobial activities of the novel clusters from this investigation has not been done as well.

## 5. Conclusions

The present study has effectively isolated and identified several streptomycetes from two different ecozones. A notable percentage of these isolates fit in uncommon species and some may represent new species although the phylogenetic tree was created from partial 16S rRNA gene. We believe the information provided will be of use to the *Streptomyces* researchers.

Expanding relevant information on *Streptomyces* communities is still needed, especially studying their dynamic range in unique ecological ecozones and their ability to produce various antibiotics to combat increasing resistance to the known agents.

## Figures and Tables

**Figure 1 ijerph-17-08749-f001:**
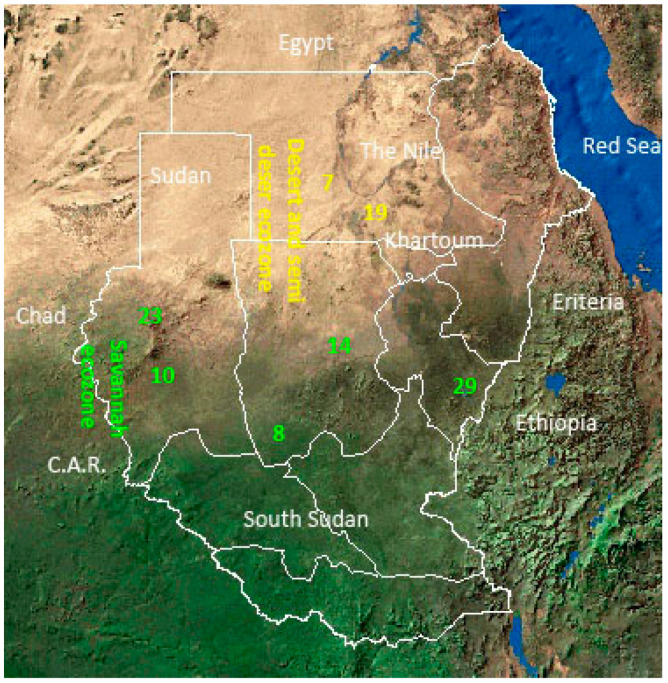
Map of Sudan showing soil collection sites and ecological zones (modified from the World Conservation Union (IUCN) (http://www.catsg.org/cheetah/04_country-information/North-African-regions/sudan/sudan-sat2.jpg).

**Figure 2 ijerph-17-08749-f002:**
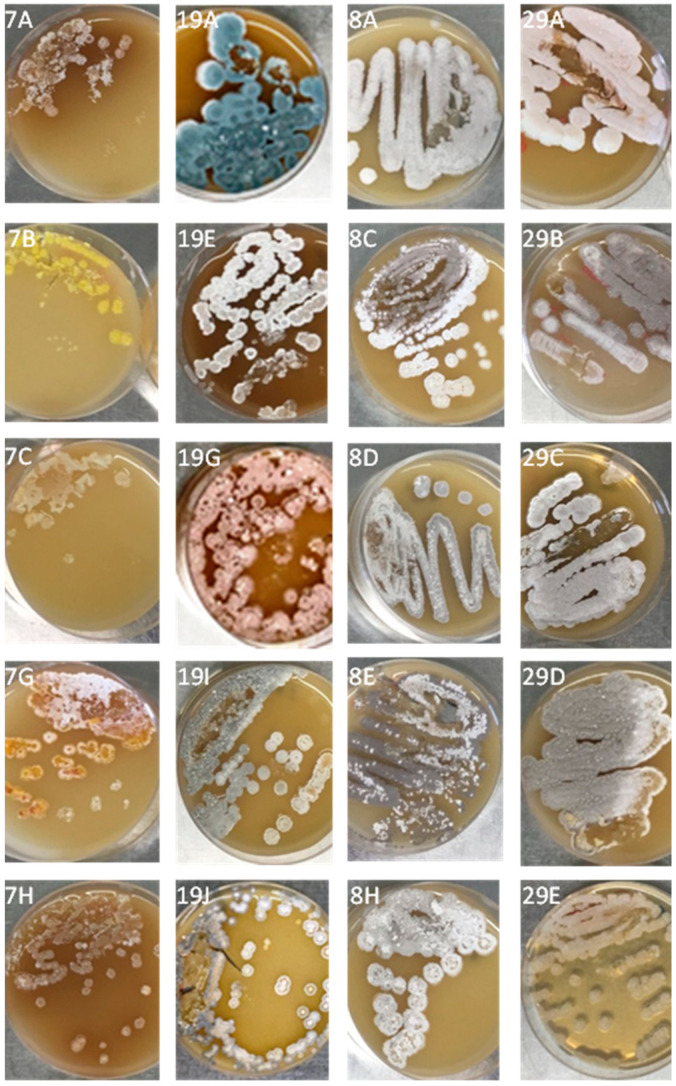
Colony morphology of *Streptomyces* spp. isolated from desert and savanna soils in Sudan using International Streptomyces Project agar number 2 (ISP2) media, incubated at 28 °C for 15 days (plate’s number indicates the soil site and letter indicates a phenotype).

**Figure 3 ijerph-17-08749-f003:**
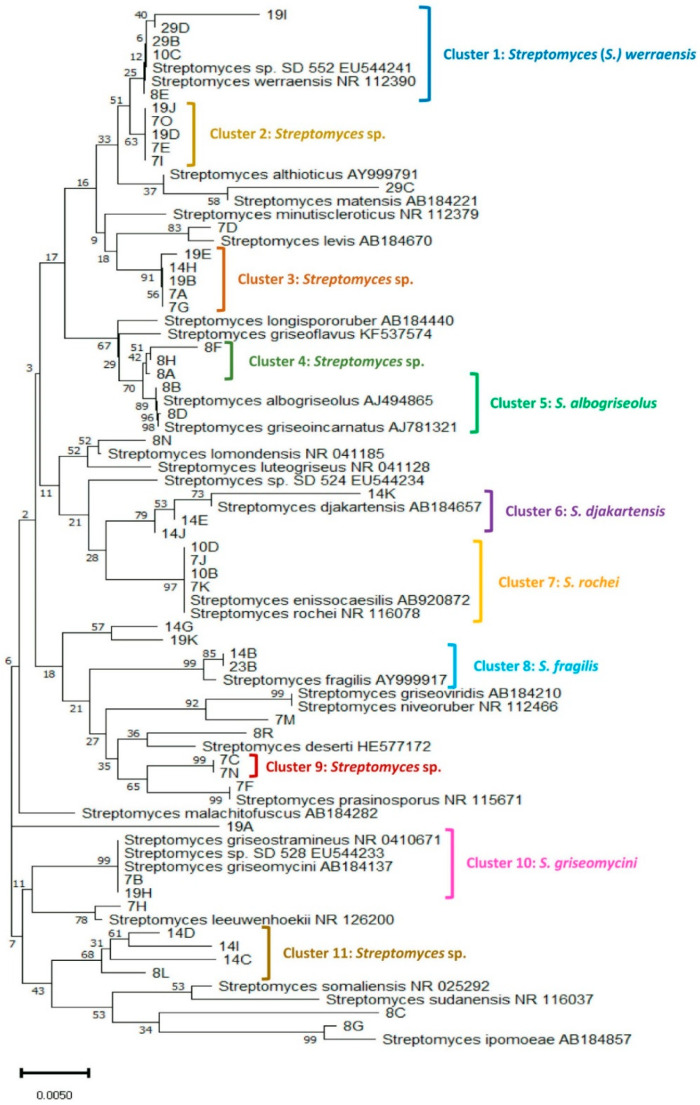
Phylogenetic relationships based on 16S rRNA sequences amongst 49 *Streptomyces* strains in relation to closely related validly described species. Evolutionary analysis was based on the Neighbor-joining method using MEGA X software [19]. The bar represents 0.005 nucleotide substitutions per alignment position; numbers above the branches are bootstrap values.

**Table 1 ijerph-17-08749-t001:** Cluster analysis of streptomycete populations isolated from desert and savanna ecozones in Sudan and identified based on 16S rRNA gene analysis.

Cluster/Species	Strain Code	Similarity Rate	GenBank Accession Number
Cluster 1: *Streptomyces* (*S.) werraensis*	29D	99.76% with *S. werraensis*	MF353969
	19I	100% with *S. werraensis*	MF353955
	29B	100% with *S. werraensis*	MF353967
	10C	100% with *S. werraensis*	MF353939
	8E	100% with *S. werraensis*	MF353990
Cluster 2: *Streptomyces* sp.	19J	99.77% with *S. werraensis*	MF356334
	7O	99.76 with *S. werraensis*	MF356359
	19D	99.78% with *S. werraensis*	MF356332
	7E	99.19% with *S. althioticus*	MF356353
	7I	99.69% with *S. althioticus*	MF356355
Cluster 3: *Streptomyces* sp.	19E	99.30% with *S. werraensis*	MF356333
	14H	99.32% with *S. griseomycini*	MF356328
	19B	99.41% with *S. griseomycini*	MF353953
	7A	98.11% with *S. griseomycini*	MF356350
	7G	99.32% with *S. griseomycini*	MF356354
Cluster 4: *Streptomyces* sp.	8F	99.34% with *S. albogriseolus*	MF356361
	8H	99.77% with *S. albogriseolus.*	MF353991
	8A	99.36% with *S. longispororuber*	MF353987
Cluster 5: *S. albogriseolus/ S. griseoincarnatus*	8B	100% with *S. albogriseolus*	MF353988
	8D	100% with *S. griseoincarnatus*	MF353989
Cluster 6: *S. djakartensis*	14E	100% with *S. djakartensis*	MF353947
	14J	100% with *S. djakartensis*	MF353948
	14K	100% with *S. djakartensis*	MF353949
Cluster 7: *S. rochei*	7K	100% with *S. rochei*	MF353986
	10B	99.70 with *S. rochei*	MF353938
	7J	98.99% with *S. rochei*	MF356356
	10D	98.83% with *S. rochei*	MF353940
Cluster 8: *Streptomyces* sp.	14B	99.42% with *S. fragilis*	MF356324
	23B	99.79% with *S. fragilis*	MF353961
Cluster 9: *Streptomyces* sp.	7C	97.69% with *S. chromofuscus*	MF356351
	7N	99.29% with *S. chromofuscus*	MF356358
Cluster 10: *S. griseomycini*	19H	100% with *S. griseomycini*	MF353954
	7B	100% with *S. griseomycini*	MF353983
Cluster 11: *Streptomyces* sp.	14D	98.88% with *S. leeuwenhoekii.*	MF356326
	14I	98.20% with *S. carpinensis*	MF356329
	14C	98.72% with *S. leeuwenhoekii*	MF356325
	8L	98.65% with *S. misionensis*	MF356363
Single-membered clusters			
*Streptomyces* sp.	7D	99.52% with *S. levis*	MF356352
*Streptomyces* sp.	8N	99.62% with *S. lomondensis*	MF356364
*Streptomyces* sp.	14G	98.51% with *S. hawaiiensis*	MF356327
*Streptomyces* sp.	19K	99.09% with *S. fimbriatus*	MF356335
*Streptomyces* sp.	7M	96.16% with *S. griseoviridis*	MF356357
*Streptomyces* sp.	8R	98.45% with *S. deserti*	MF356365
*S. prasinosporus*	7F	100% with *S. prasinosporus*	MF353984
*Streptomyces* sp.	19A	98.95% with *S. malachitofuscus*	MF356331
*Streptomyces* sp.	7H	99.88% with *S. leeuwenhoekii.*	MF353985
*Streptomyces* sp.	8C	98.93% with *S. capitiformicae*	MF356360
*Streptomyces* sp.	8G	99.61% with *S. Ipomoeae*	MF356362
*Streptomyces* sp.	29C	98.77% with *S. matensis*	MF353968

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
