# Peer review of "Isolation and Identification of *Streptomyces* spp. from Desert and Savanna Soils in Sudan"

_ijerph, 2020, doi:10.3390/ijerph17238749_

Round 1
Reviewer 1 Report
This is a manuscript to isolate and identify streptomycetes from in desert and savanna zones in Sudan and to assess the species diversity in the two zones using 16S rRNA gene analysis.The manuscript needs to be carefully revised.
- Firstly, sequences used in 16S rRNA gene sequence similarity calculation must be as accurate as possible.
As a minimum, (i) both strands should be sequenced and (ii) 16S rRNA gene sequences should be complete. However, in this study. the authors obtained 16S rRNA gene sequences that were too shorter(<1000bp) to assure the quality of 16S sequences.
- The obtained 16S rRNA gene sequences should be compared with available 16S rRNA gene sequences of validly published species on EzBioCloud server (www.ezbiocloud.net) to determine an approximate phylogenetic affiliation. However, the authors have not carried out it until now.
For example, The EzBioCloud analysis of the 16S rRNA gene sequence of strain 8A revealed the strain shared the highest sequence similarity to Streptomyces longispororuberNBRC 13488T (99.36%), but that to Streptomyces albogriseolusNRRL B-1305T (98.74%) .
- If so, the strains diversity shouldbe reassessed according to the EzBioCloud analysis.
Author Response
Response to Reviewer’s comments and suggestions
- This is a manuscript to isolate and identify streptomycetes from in desert and savanna zones in Sudan and to assess the species diversity in the two zones using 16S rRNA gene analysis.The manuscript needs to be carefully revised.
- Firstly, sequences used in 16S rRNA gene sequence similarity calculation must be as accurate as possible. As a minimum, (i) both strands should be sequenced and (ii) 16S rRNA gene sequences should be complete. However, in this study. The authors obtained 16S rRNA gene sequences that were too shorter(<1000bp) to assure the quality of 16S sequences.
Author’s Response:
We agree the sequnce was shorter than the ideal ca. 1400bp.
We used the good quality sequnces obtianed from of the strands, now we aligned the best of thetwo strandst that was deposited inGenBank and used in alignment in the MEGA software.
- The obtained 16S rRNA gene sequences should be compared with available 16S rRNA gene sequences of validly published species on EzBioCloud server (www.ezbiocloud.net) to determine an approximate phylogenetic affiliation. However, the authors have not carried out it until now. For example, The EzBioCloud analysis of the 16S rRNA gene sequence of strain 8A revealed the strain shared the highest sequence similarity to Streptomyces longispororuber NBRC 13488T (99.36%), but that to Streptomyces albogriseolusNRRL B-1305T (98.74%) .
- If so, the strains diversity should be reassessed according to the EzBioCloud analysis.
Response:
EzBioCloud analysis was used and accordingly Table (1) was rewritten. Also Figure1 was changes.
We agree, some changes have been resulted in the identification.
Thanks for this advice.
Reviewer 2 Report
This paper is sound in and of itself, but it presents far too small a data set to inform readers of the diversity of Streptomyces across such a vast area.
Author Response
Response to Reviewer’s comments and suggestions
This paper is sound in and of itself, but it presents far too small a data set to inform readers of the diversity of Streptomyces across such a vast area.
Response:
The paper has limited collection of samples and also from limited area within the compared two ecozones. We think as a short communication could give an idea about the diversity. The word diversity is now deleted from the title and instead the title bears “identification” which we think is suitable for the objective of this study
Reviewer 3 Report
To Authors,
The manuscript has been improved compared to the previous version. However, this needs more corrections to be published.
I recognised the change in the title which is more appropriate for the target of this investigation. Along with the manuscript, however, you can still see the focus on diversity such as in the discussion where most of the references cited come from studies characterising diversity. To my understanding, the target here is the identification of Stremtomycetes and the discussion should provide emphasis on why there are different types of these microbes in one or another ecozone be studied. What makes these microbes to prefer for one or another and references should be used to support this discussion-. At the methodological level, the discussion should be done with studies of similar characteristics.
There are a few English issues along with the manuscript and also issues about formality in the language applied. e.g. Line 51 - good deal. Soil descriptions such as Arenosols - stabilized sand dune with silt or clay in Line 118.
The methodology still lacks if details. Firstly, the workflow is a bit confusing - e.g. not so clear the order from soil to isolates to cultures to sequencing. Perhaps, the addition of a figure or scheme and map showing locations from ecozones could help.
References are missing - e.g. Line 44. Microbes affected by environmental factors. examples??
Line 52. Studies in Sudan with these findings. Name at least one.
Line 126 pick spot technique - any reference?
Discussion in general - as mentioned before, this should focus on describing results according to the products of this research - Half of the discussion describes what should be as introductions or make reference to studies that don't have the same characteristics.
At least, it would be good to know what would be the potential of the spp found based on a literature review.
In the method, the bioinformatic description lacks details - e.g. did you use a filtering process for the sequences etc.
The conclusion is weak - you need to emphasise the applicability and uniqueness of these findings. What's a step forward in this area after this research is generally described but mention something that would be a real contribution in this area. You can take the unique ones were found after this study and provide more feedback about those, for example.
These can be extended to the methods applied, - e.g. you mention how the species found from the isolates were represented in the sequencing performance but how many were not, for example?
Author Response
Response to Reviewer’s comments and suggestions
- The manuscript has been improved compared to the previous version. However, this needs more corrections to be published.
- I recognized the change in the title which is more appropriate for the target of this investigation. Along with the manuscript, however, you can still see the focus on diversity such as in the discussion where most of the references cited come from studies characterising diversity. To my understanding, the target here is the identification of Stremtomycetes and the discussion should provide emphasis on why there are different types of these microbes in one or another ecozone be studied. What makes these microbes to prefer for one or another and references should be used to support this discussion-. At the methodological level, the discussion should be done with studies of similar characteristics.
Response:
This issue was addressed.
Lines 193 to 208 reflect our view in relation to literature regarding the identification methods in actinomycetes.
- There are a few English issues along with the manuscript and also issues about formality in the language applied. e.g. Line 51 - good deal. Soil descriptions such as Arenosols - stabilized sand dune with silt or clay in Line 118.
Response:
English has been revised and with help from the Academic Editor Notes, which we have addreded. Newly added correction is Track-changed.
- The methodology still lacks if details. Firstly, the workflow is a bit confusing - e.g. not so clear the order from soil to isolates to cultures to sequencing. Perhaps, the addition of a figure or scheme and map showing locations from ecozones could help.
Response:
We agree a map would be useful. Figure 1 now is a map.
We think the following flow of methods is logical and that what we have actually did.
- Study samples and geographical locations
- Pretreatment of soil samples
- Isolation of Streptomyces
- Characterization of soil isolates
- Molecular identification of soil isolates
- References are missing - e.g. Line 44. Microbes affected by environmental factors. examples??
Line 52. Studies in Sudan with these findings. Name at least one.
Line 126 pick spot technique - any reference?
Response:
A reference is provided.
- Discussion in general - as mentioned before, this should focus on describing results according to the products of this research - Half of the discussion describes what should be as introductions or make reference to studies that don't have the same characteristics.
Response:
Lines 190-205 address this issue. We discuss the the value of different identification methods from literature in comparison to our chosen method
- At least, it would be good to know what would be the potential of the spp found based on a literature review.
Response:
We have indicated the potential use of these isolates and compared with some ppapers in the literature
e.g. Lines. 206-227.
- In the method, the bioinformatic description lacks details - e.g. did you use a filtering process for the sequences etc.
Response:
This part was re-written Lines 113-118:
Description of handling the generated raw sequence in blast and later with EZbioCloud to establsih a quick phylogentic position, Sequences were then assembld, corrected, trimmed using Chromas and analysied using MEGA X. (No filteringhave been done).
We believe such detail is enough for a short ommuction!
- .The conclusion is weak - you need to emphasise the applicability and uniqueness of these findings. What's a step forward in this area after this research is generally described but mention something that would be a real contribution in this area. You can take the unique ones were found after this study and provide more feedback about those, for example.
Response:
Amended, Line2231-234
- These can be extended to the methods applied, - e.g. you mention how the species found from the isolates were represented in the sequencing performance but how many were not, for example?
Response:
In this paper all selected phenotypes haven been sequencd, In fact we have large collection of isolates but the slected ones are complete in term of identification but not full descriptions..
Round 2
Reviewer 2 Report
I appreciated the change of emphasis in the title to state "Isolation and identification" rather than "Diversity of...". This does bring the findings in line with the claim the title makes.
This manuscript is a resubmission of an earlier submission. The following is a list of the peer review reports and author responses from that submission.
Round 1
Reviewer 1 Report
This is a manuscript to isolate and identify streptomycetes from in desert and savanna zones in Sudan and to assess the species diversity in the two zones using 16S rRNA gene analysis.Therefore, I would like to suggest the authors that the obtained 16S rRNA gene sequences are compared with available 16S rRNA gene sequences of validly published species on EzBioCloud server (www.ezbiocloud.net) (Yoon et al. 2017) to determine an approximate phylogenetic affiliation. If so, the species diversity must be reassessed.
The following is the reference:
Yoon SH, Ha SM, Kwon S, Lim J, Kim Y et al. Introducing EzBioCloud: a taxonomically United database of 16S rRNA gene sequences and whole-genome assemblies. Int J Syst Evol Microbiol 2017;67:1613–1617.
Reviewer 2 Report
The manuscript by Hamid et al reports on work to “isolate and identify streptomycetes from in [sic] desert and savanna zones in Sudan and to find out the species diversity in the two zones using 16SrRNA gene analysis.” The paper goes on to outline isolation and phylogenetic characterization of 49 Streptomyces obtained from multiple desert and savanna locations in Sudan. The methods used are appropriate and the data are well presented. The weakness of the reported work lies in the approach to the question. Bacterial diversity in soils is known to be high, and that includes the diversity of actinomycetes and Streptomyces specifically, and the same is true for the culturable diversity. By obtaining one or a few isolates per sample site, a random and most likely very partial representation of the true Streptomyces diversity would be obtained. The approach taken implies that the results cannot be held up as a representation of the diversity of Streptomyces in this vast tract of land.
The isolates obtained from this ecozone, probably not sampled extensively before, may be novel and have unique properties. A thorough phenotypical characterization of these possibly unique isolates could reveal synthesis of novel compounds of medicinal or industrial use. The authors are encouraged to investigate these isolates for synthesis of compounds of medicinal value such as antibacterial substances, anti-fungal compounds. Access to cell culture facilities could also enable screening for other classes of medicinal compounds.
Specific comments:
- The genus Streptomyces has an extraordinarily large number of species, and the 16SrRNA genes are less variable across members of the genus than average, so that some species have almost identical 16S sequences. To obtain a clear phylogeny of the isolates within the genus, a Multi-Locus Sequence Typing (MLST) approach is advised, for example as described by Guo, Y., Zheng, W., Rong, X., & Huang, Y. (2008). A multilocus phylogeny of the Streptomyces griseus 16S rRNA gene clade: use of multilocus sequence analysis for streptomycete systematics. Int J Syst Evol Microbiol, 58(Pt 1), 149-159. doi:10.1099/ijs.0.65224-0
- How did the authors obtain complete 16SrRNA sequences – only by sequencing from the 27f and 1492r primer sites? Sanger chain termination sequencing very rarely yields reads that are reliable for over 800 bases, so internal primers are usually employed to ensure reliable reads.
- The are no details given on how the phylogenetic tree was generated – other than stating use of Mega (line 118). There are many approaches – e.g. Maximum Likelihood. Also, the number of runs to arrive at the bootstrap values should be defined – 1,000?
Reviewer 3 Report
The manuscript entitled “Diversity of Streptomyces spp. in desert and savanna soils in Sudan” described isolation of streptmycetes strains from desert and savanna soils in Sudan, their identification are carried out. These results are of some interest to researchers in microbial ecology. But this research is preliminarily. The authors should further study the ecological characteristics or metabolic characteristics of Streptomyces in order to show the environmental adaptation mechanism of Streptomyces, or to study the secondary metabolite abundance.
Reviewer 4 Report
To authors,
The manuscript presents an investigation aiming to isolate, identify and estimate the diversity of streptomycetes from desert and savanna zones in Sudan.
The main novelty relies upon the discovery of new streptomycetes clusters from soils in these zones since they may have potential applications as antibacterials, antifungal or antitumoral. It would be interesting as a scientific contribution to see the results of the tested antibacterial activity (screening) of the novel clusters from this investigation, for example.
Major concerns are in the reported results, discussion and conclusions.
Results can be improved as should give more references to other studies - e.g contrasting or similarities to other findings to elucidate an explanation responding to why these microbes are in these environments. This helps for future investigations were to explore according to these microbes drivers and preferences. There are assumptions of site-specific results but it is not clear to me whether this can be generalised for all different types of soils - e.g. Savanna has Arenosols and Vertisols. I don't see reported results pointing out such differences. Results could show the effect of the environmental differences among zones - e.g. specific preferences for the unique taxa.
The title of the paper itself makes reference to 'Diversity'. There are not diversity measurements shown in the result sections itself. It would be good to add graphic representations and statistical support for diversity analysis. Otherwise, diversity is not analysed but only identification and perhaps a different title should be considered.
The discussion uses references from the study carried in the Atacama Desert in Chile. It is mentioned similar results to Savanna soils in this study based on abundance, reductions of diversity, soil relative humidity but none of those parameters was presented in this study. Maybe a deeper discussion on streptomycetes in general in relations to different environments can be added.
Lines 182-186 are mostly part of an introduction than discussion since none of those points was tested yet in this research. The research findings cannot be a platform for new research field but for potential application of those new species yes it is.
Conclusions should add details on what else would be needed to make use of this new knowledge or the remaining questions. It states 'notable percentage of the isolates fit uncommon species' - this should be discussed deeper in the discussion section and be paralleled to other findings fitting uncommon species even in other taxa.
Minor concerns are English style issues along with the manuscript that should be improved. e.g. 170-171 - it sounds like an unfinished sentence.
In general, there is repetitive information along with the manuscript that can be avoided and instead be enriched by other studies references and/or more analysis from this study.